# The ncRNAs Involved in the Regulation of Abiotic Stress-Induced Anthocyanin Biosynthesis in Plants

**DOI:** 10.3390/antiox13010055

**Published:** 2023-12-28

**Authors:** Bo Zhou, Baojiang Zheng, Weilin Wu

**Affiliations:** 1College of Life Science, Northeast Forestry University, Harbin 150040, China; zhengbaojiang@nefu.edu.cn; 2Agricultural College, Yanbian University, Yanji 133002, China

**Keywords:** anthocyanin biosynthesis, ncRNA, environmental regulation, abiotic stress, regulatory network

## Abstract

Plants have evolved complicated defense and adaptive systems to grow in various abiotic stress environments such as drought, cold, and salinity. Anthocyanins belong to the secondary metabolites of flavonoids with strong antioxidant activity in response to various abiotic stress and enhance stress tolerance. Anthocyanin accumulation often accompanies the resistance to abiotic stress in plants to scavenge reactive oxygen species (ROS). Recent research evidence showed that many regulatory pathways such as osmoregulation, antioxidant response, plant hormone response, photosynthesis, and respiration regulation are involved in plant adaption to stress. However, the molecular regulatory mechanisms involved in controlling anthocyanin biosynthesis in relation to abiotic stress response have remained obscure. Here, we summarize the current research progress of specific regulators including small RNAs, and lncRNAs involved in the molecular regulation of abiotic stress-induced anthocyanin biosynthesis. In addition, an integrated regulatory network of anthocyanin biosynthesis controlled by microRNAs (miRNAs), long non-coding RNAs (lncRNAs), transcription factors, and stress response factors is also discussed. Understanding molecular mechanisms of anthocyanin biosynthesis for ROS scavenging in various abiotic stress responses will benefit us for resistance breeding in crop plants.

## 1. Introduction

With population growth and environmental deterioration, plants face more and more abiotic stresses. As sessile organisms, they are exposed to various abiotic stresses during their growth and development process, and they have also evolved sophisticated tolerance, resistance, or avoidance mechanisms to overcome stress such as low and high temperature, drought, and salinity [1,2]. Under abiotic stress, the stress stimuli are perceived by receptors or signal factors and transduced to downstream transcription factors through messengers like calcium (Ca^2+^), nitric oxide (NO), sugars, abscisic acid (ABA), brassinosteroids (BRs), ethylene, jasmonates (JA), salicylic acid (SA), and auxins in plants [3]. Under abiotic stresses, plants usually generate and accumulate secondary metabolites including anthocyanins, flavones, flavonols, lignin, alkaloids, and terpenoids to protect cells from damage [4,5]. Studies have shown that anthocyanins accumulate in plants under drought, cold, and salinity stress and the higher anthocyanin content increases the tolerance of plants to adapt to harmful environmental stress [6,7,8].

Anthocyanins are a class of water-soluble flavonoids present in fruits, flowers, and vegetative organs of plants which have antioxidant activities in protecting plants under abiotic stress. Anthocyanidins are synthesized on the cytoplasmic surface of the endoplasmic reticulum and further undergo various modifications such as methylation, glycosylation, hydroxylation, and acylation in the endoplasmic reticulum. Afterward, anthocyanins (the forms of anthocyanidin glycosides and acylated anthocyanins) enter the vacuoles to store and accumulate with the assistance of transporters and transport vesicles [9]. Anthocyanins not only determine the blue, red, and purple pigments of plants for attracting pollinators but also play roles in various biotic and abiotic stresses. In addition to physiological roles for plants, anthocyanins also have potential benefits for human health, such as decreasing the risk of heart disease, diabetes, cardiovascular disease, and metabolic diseases [10,11,12,13]. The anthocyanin biosynthetic pathway has been elucidated and most of the regulatory genes involved in anthocyanin biosynthesis have been identified [14,15,16]. The enzymes involved in anthocyanin biosynthesis include CHS (chalcone synthase), CHI (chalcone isomerase), F3H (flavanone 3-hydroxylase), F3′H (flavonoid 3′-hydroxylase), F3′5′H (flavonoid 3′,5′-hydroxylase) which correspond to early biosynthetic genes (EBGs), and DFR (dihydroflavonol 4-reductase), ANS (anthocyanin synthase), OMT (O-methyltransferase) and UFGT (UDP flavonoid glucosyltransferase) which correspond to late biosynthetic genes (LBGs) [17]. Moreover, the anthocyanin biosynthesis is regulated by the MYB–bHLH–WD40 (MBW) protein complex, which is composed of MYB, bHLH transcription factors, and a WD40 protein [18,19]. In addition to the MBW complex, PybHLH3-PyMYB114-PyERF3 [20] transcription complex in pear and WRKY [21,22], NAC [23], MADS [24], HY5 [25], BBX [26], bZIP [27], SPL [28] regulatory factors in various fruit crops are also involved in the regulation of anthocyanin biosynthesis (Figure 1). Recently, a great number of studies have revealed that anthocyanins increasingly accumulate when plants are under environmental stress. In *Arabidopsis*, the low nitrogen (N)-induced anthocyanin accumulation plays a substantial role in plant tolerance to low N stress [29]. In addition, the flavonoid biosynthesis and accumulation in *Arabidopsis* improves salt resistance under salt stress [30]. In *Cymbidium* hybrid flowers, anthocyanin pigmentation has been demonstrated to be organ-specific and temperature-dependent synthesized [31]. Moreover, anthocyanin accumulation is related to salt stress response in MdZAT5-overexpressing apple Calli and *Arabidopsis* [32]. Furthermore, the anthocyanin accumulation in *AN1*-overexpressing tobacco plants has a higher drought tolerance compared to the wild-type plants [33]. Also, the overexpression of *UGTs* enhanced plant tolerance to low temperatures, drought, and salt stresses by modulating the anthocyanin accumulation [34]. Under abiotic stresses, a reduction in electron transport in the Calvin cycle and a higher electron leakage during photosynthesis in the Mehler reaction of cells lead to the plants producing extensive reactive oxygen species (ROS) including O^2−^, H_2_O_2_, OH^-^, and ^1^O_2_ [35], which cause oxidative damage to plants and can be used as signaling molecules to activate stress-tolerance mechanisms. Anthocyanin can directly scavenge active oxygen species, such as singlet oxygen, superoxide, hydrogen peroxide, hydroxyl, and peroxyl radicals [36]. Consequently, the anthocyanins accumulate in plants after ROS signals induce the transcription of anthocyanin biosynthesis pathway genes to scavenge excess ROS and avoid oxidative damage [37,38]. Therefore, ROS as signal factors in the response to plant abiotic stresses activate the transcription of anthocyanin biosynthetic genes to produce anthocyanins for stress tolerance. 

In general, non-coding RNAs (ncRNAs) have been classified into housekeeping ncRNAs (constitutive expression) such as ribosomal RNA (rRNA), transfer RNA (tRNA), small nuclear RNAs (snRNA), small nucleolar RNAs (snoRNAs), TR (telomerase RNA) and regulatory RNAs such as microRNAs (miRNA), endogenous-small-interfering RNA (endo-siRNA), repeat-derived RNA (rasiRNA), Piwi-interacting RNA (piRNA), enhancer-derived RNA (eRNA), promoter-associated RNAs (PATs), long non-coding RNA (lncRNA) [39]. The biogenesis, targeting action, and function of these classes of ncRNAs have been explored and reviewed in plant development [40,41,42,43,44], whereas the regulatory RNAs (miRNAs and lncRNAs) in the regulatory pathway of anthocyanin biosynthesis in abiotic stress response still need to be explored. miRNAs and lncRNAs originate in intergenic or intronic regions of chromosomal DNA and regulate the expression of growth and development, biotic and abiotic stress response-related genes [43]. They are involved in chromatin modification, epigenetic regulation, genomic imprinting, transcriptional control, and pre- or post-translational mRNA processing in diverse biological processes in plants [45,46]. In sea buckthorn, two lncRNAs *LNC1* and *LNC2* can act as endogenous target mimics of miR156a and miR828a to regulate the abundance of *SPL9* and *MYB114*, respectively, which affect the anthocyanin biosynthesis in fruit [47]. Additionally, in apples, lncRNA *MdLNC610* upregulates the expression of the ethylene biosynthesis gene *MdACO1* and participates in the regulation of high-light-induced anthocyanin production [48]. Another lncRNA *MdLNC499* which is regulated by MdWRKY1 induces the expression of *MdERF109* and participates in regulating the accumulation of anthocyanins in light-induced apple fruit coloration [49]. Moreover, the anthocyanin-rich tomato genotype LA1996 displays superior tolerance to salinity and drought stress [7]. Drought, salinity, and temperature (heat, cold, chilling, and freezing) are major abiotic stresses affecting the growth and development of plants. Many signal transduction regulators and key transcription factors involved in the complicated regulatory network of abiotic stresses have been identified. Recent studies have revealed more and more miRNAs and lncRNAs related to the regulatory pathway of anthocyanin biosynthesis under various abiotic stress environments in plants. In this review, we focus on the current advances of miRNAs and lncRNAs involved in the regulation of anthocyanin biosynthesis induced by abiotic stress and their roles in abiotic stress tolerance. 

## 2. Environmental Stress-Induced Anthocyanin Accumulation in Plants

### 2.1. The Pathway of Transcription Factors and ncRNAs Involved in Low Temperature Stress Response

Low temperature is one of the major abiotic stresses that greatly reduces crop yield. The ability of plants to tolerate adverse environments is known to affect plant survival and geographical distribution, for example, plants growing in temperate zones are less sensitive to cold than those from tropical/subtropical regions, such as rice, maize, cotton, and tomatoes, which cannot adapt to cold environments [50]. Cold stress can be classified into chilling stress (0–15 °C) and freezing stress (<0 °C), which can cause injuries to the plant. However, some plants such as winter wheat (*Triticum aestivum*), rye (*Secale cereale* L.), barley (*Hordeum vulgare*), and oat (*Avena sativa*) have evolved sophisticated cold acclimation mechanisms to encounter low temperatures that improve plant freezing tolerance [51]. 

Currently, the cold signal has been reported to be perceived by cellular membranes, calcium (Ca^2+^) channels, and COLD1 (CHILLING TOLERANCE DIVERGENCE 1), which encodes a G-protein signaling regulator in *Oryza sativa* [52]. Afterwards, the cold signal is transduced to the C-REPEAT BINDING FACTOR/DRE BINDING FACTOR1 (CBF/DREB1)-dependent regulatory pathway for plant responses to cold stress. Additionally, the circadian clock, photoperiod, and light signaling are involved in regulating the network [52]. CBF proteins recognize the conserved CCGAC sequence of *CRT/DRE* cis-element in the promoters of a subset of cold-regulated (*COR*) genes [53]. The cold-regulated genes including the *COR*, low-temperature induced (*LTI*), responsive to desiccation (*RD*), and early dehydration-inducible (*ERD*) genes can be activated to increase freezing tolerance through the regulation of the CBF/DREB1 transcription factors in the cold acclimation of plants [51]. During the cold signal transduction, ICE1 (INDUCER OF CBF EXPRESSION 1) and its homolog ICE2 positively regulate *CBF* expression, and the modifications of post-translation in ICE1 such as ubiquitination, sumoylation, and phosphorylation attenuate the activity and stability of ICE1 to affect the expression abundance of *CBF* [51,52] (Figure 2). Moreover, overexpression of *HY5* or *MYB15* increases the expression of *CBF* and cold tolerance in tomatoes, and loss-of-function mutations *hy5* or *myb15* decrease the levels of *CBF* transcripts [54]. 

Furthermore, CBF can interact with PHYTOCHROME-INTERACTING FACTOR 3 (PIF3) under cold stress attenuating the mutually assured destruction of PIF3-phyB to stabilize PHYTOCHROME B (phyB) and positively regulate freezing tolerance by modulating the expression of stress-responsive genes in *Arabidopsis* [55]. In addition, PIF3 can be ubiquitinated by LRBs (LIGHT-RESPONSE BRIC-A-BRACK/TRAMTRACK/BROAD) for degradation after interacting with PhyB to be phosphorylated [56,57,58,59]. Nevertheless, PIF3 regulates anthocyanin biosynthesis in an HY5-dependent manner with both factors directly binding anthocyanin biosynthetic gene promoters [60]. In addition, CBF1 in *Arabidopsis* can also control the expression of the two glycosyltransferase genes (*UGT79B2* and *UGT79B3*) involved in modulating anthocyanin metabolic pathways to improve cold stress tolerance [61]. 

Moreover, CBFs in eggplants interact with SmMYB113, a key regulator of anthocyanin biosynthesis, to upregulate the expression of *CHS* and *DFR* with a SmMYB113-dependent pathway to improve the contents of anthocyanin. Overexpression of *SmCBF2* and *SmCBF3* in *Arabidopsis* also enhances the anthocyanin accumulation under cold conditions [62]. Additionally, MdMYB23 directly activates the expression of *MdCBF1* and *MdCBF2* to enhance the cold tolerance of apple plants (*Malus domestica*) at low temperatures. MdMYB23 also regulates the transcription of *MdANR* and promotes the biosynthesis of proanthocyanidins to scavenge ROS for cold stress tolerance [63]. Conversely, the MdMYB15L repressor binds to the promoter of *MdCBF2* and inhibits the expression of *MdCBF2*, which also competitively binds with MdbHLH33 and reduces MdbHLH33-induced anthocyanin accumulation, thus decreasing the cold tolerance of apple at low temperatures [64]. 

Moreover, the NAC (NAM, ATAF1/2, and CUC2) transcription factor MdNAC104 in apples also promotes the expression of *MdCBF1* and *MdCBF3*, the anthocyanin synthesis-related genes such as *MdCHS-b*, *MdCHI-a*, *MdF3H-a* and *MdANS-b* and the antioxidant enzyme-encoding genes *MdFSD2* and *MdPRXR1.1* under cold stress [65]. Furthermore, *MdMYB308L*, overexpression of which improved cold tolerance and increased anthocyanin accumulation, interacts with MdbHLH33 to regulate the expression of *MdCBF2* and *MdDFR*. However, an apple RING E3 ubiquitin ligase MdMIEL1 promotes the degradation of MdMYB308L and negatively regulates cold tolerance and anthocyanin accumulation [8] (Figure 2). As a result, both CBFs and PIF3 take dual regulation roles in cold stress response and anthocyanin accumulation in plants. 

Apart from the CBF-dependent regulatory pathway, a number of genes have been identified to regulate plant cold responses independently of the CBF pathway (Figure 3). In *Arabidopsis*, cold stress induces the expression of *HY5* (ELONGATED HYPOCOTYL 5) and positively regulates the expression of *COR* genes and cold acclimation via a CBF-independent pathway [66]. Moreover, HY5 has been determined to regulate anthocyanin biosynthesis by inducing the transcriptional activation of the MYB75/PAP1 transcription factor in *Arabidopsis* [67] and regulating the expression of anthocyanin biosynthetic genes (*CHS*, *CHI*, and *F3H*) in tomato under cold stress [68]. Additionally, ROS1 (REPRESSOR OF SILENCING 1), one of the DNA glycosylase DEMETER (DME) family members promotes low temperature-induced anthocyanin accumulation in apple (*Malus domestica*) by demethylating the promoters of anthocyanin-associated genes (*MdCHS*, *MdCHI*, *MdF3’H*, *MdANS*, *MdUFGT*, and *MdMYB10*) [69]. 

Low temperature is a serious abiotic stressor that induces ROS (reactive oxygen species), which participate in stress signaling as signaling molecules through the protein kinases pathway to activate anthocyanin regulatory transcription factors such as MYB, bHLH, and WD40 [70,71]. The accumulation of anthocyanins in plant leaves at low temperatures can decrease the level of oxidative damage and increase the photosynthetic rate [72]. In *Mikania micrantha*, the accumulation of anthocyanins in leaves and stems could effectively eliminate ROS under low-temperature stress and improve its adaptability to low-temperature environments during winter [73]. The anthocyanins scavenging ROS and maintaining osmotic balance to increase abiotic stress tolerance has also been proved using anthocyanin-containing leaves compared with anthocyanin-deficient leaves in sweet basil (*Ocimum basilicum*) and *Arabidopsis* [71,74]. Also, cold-induced transcription and phosphorylation of bHLH activate the expression of *MdDFR*, *MdUFGT*, and *MdMYB1* to regulate anthocyanin accumulation and fruit coloration in apples [75]. Additionally, low temperature induces *MdMYB2* and further activates the expression of *MdSIZ1* (SAP AND MIZ1 DOMAIN-CONTAINING LIGASE1), which is SUMO E3 ligase that mediates the sumoylation of MdMYB1 and promotes anthocyanin accumulation [76,77] (Figure 3). Furthermore, BrMYB2 and BrTT8 play important roles in co-activating the anthocyanin structural genes in purple-head Chinese cabbage after low-temperature induction [78]. In Chinese cabbage, low temperature also induces the expression levels of *MYB12*, *MYB75*, *MYB111*, *MYB113*, and *MYB114* leading to the increased expression of structural genes for anthocyanin accumulation [79]. On the contrary, in strawberry fruit, low temperature inhibits anthocyanin accumulation by activating FvMAPK3-induced phosphorylation of FvMYB10 and degradation of chalcone synthase 1 [80]. The negative regulatory pathway of anthocyanin biosynthesis might maintain the balance of anthocyanin accumulation under cold stress. Therefore, the CBFs-independent pathway is also involved in the regulation of cold stress response and anthocyanin accumulation in plants. 

Non-coding RNAs also play crucial roles in the plant’s response to cold stress. An overexpression of *MIR156* in tobacco leaves for transient expression assay showed higher anthocyanin levels and an enhanced cold stress tolerance than that in the control. Additionally, the lower extent of H_2_O_2_ accumulation and the higher expression of early responsive dehydration (ERD) gene family, *ERD10B*, *ERD10C,* and *ERD10D*, which are downstream genes in the ICE-CBF-COR pathway, were detected under cold stress [81]. *SPL9*, one of the miR156 targets, has been reported to inhibit the expression of anthocyanin synthesis genes by competitively interacting with PAP1 (production of the anthocyanin pigments1), the component of the MYB–BHLH–WD40 complex, resulting in the interference with the regulation of anthocyanin accumulation [82]. The negative regulation of *SPL9* targeted by miR156 caused the accumulation of anthocyanins in the *MIR156*-overexpressing tobacco leaves preventing the overproduction and accumulation of ROS in tobacco leaves. The miR828/858-MYBs factors are also related to regulating needle discoloration of *C. fortunei* in cold winters [83]. Another miRNA, miR319, has highly conserved sequences in rice (*Oryza sativa* L.) and *Arabidopsis*. OsmiR319 positively regulates cold tolerance by targeting *OsPCF6* and *OsTCP21* (TEOSINTE BRANCHED/CYCLOIDEA/PCF) in rice [84] and AtmiR319 negatively regulates anthocyanin biosynthesis by targeting *AtTCP3* which can interact with R2R3-MYB in *Arabidopsis* [85]. In sugarcane plants, miR319 is induced by cold (4 °C) stress, whereas the expression of *PCF5* and *PCF6* targeted by miR319 is down-regulated under cold treatment [86]. Apparently, the miR319-*TCP* mediated a complex gene expression regulatory network to exhibit multi-functions in cold tolerance and anthocyanin biosynthesis with species-specific. features.

Furthermore, miR397 overexpression in *Arabidopsis* improves tolerance to cold stress by inducing high levels of *CBF* and *COR* by targeting *LACs* (Laccase) and *CKB3* (Casein Kinase II Subunit Beta 3) in the cold signaling pathway [87]. Whereas the lncRNA, FRILAIR (FRUIT RIPENING-RELATED LONG INTERGENIC RNA) acts as a noncanonical target mimic of miR397 to modulate the expression of *LAC11a* (encoding a putative laccase-11-like protein), which promotes expressions of anthocyanin biosynthetic genes in the strawberry fruit ripening process [88]. Similar to miR397, miR408 is highly conserved among different species, which can also target Laccase (*LAC12* and *LAC13*) involved in light and copper signaling in *Arabidopsis* and *amiR408* (artificial miRNA silences MIR408) seedlings display reduced anthocyanin content [89]. In *Oryza sativa*, OsmiR408 is induced by cold stress, and the tolerance to cold stress is also improved in OsmiR408 overexpressing seedlings [90]. Another ncRNA, miR398, which can interact with lncR9A, lncR117, and lncR616 in winter wheat, targets and regulates *CSD1* (Cu-Zn-type superoxide dismutase 1) expression to improve cold resistance [91]. In peanuts, the AhmiR398 also regulates *AhCHS* to be involved in the anthocyanin biosynthesis [92]. 

In addition, MdLNC499 can target *MdERF109* to regulate light-induced anthocyanin accumulation in apple fruit [49] and the ERF109 in trifoliate orange also contributes to cold tolerance by regulating *Prx1* involved in the antioxidative process [93]. Another lncRNA, MdLNC610, upregulates the expression of *MdACO1* and participates in high-light-induced ethylene and anthocyanin biosynthesis [48]. Moreover, in apples, *MdARF17* (AUXIN RESPONSE FACTOR 17) targeted by miR160 plays a positive role in apple freezing tolerance by promoting the expression of *MdWRKY33* which regulates the expression of the cold-responsive genes and ROS-related genes in response to cold [94]. Furthermore, cold-induced lncRNA 1 (CIL1) in *Arabidopsis* enhances cold stress tolerance by regulating the expression of multiple stress-related genes and activates the endogenous reactive oxygen species (ROS) metabolism pathway [95] (Table 1). Overall, HY5 and transcription factors respond to light and low temperature signals suggesting the coordinate pathway in the regulation of anthocyanin pigmentation in plants.

### 2.2. High-Temperature Stress Response Factors

Floral pigmentation has been reported to respond differently to changes in temperature, and plants with exposed anthers increased in pigmentation with temperature increasing whereas plants with concealed anthers declined in pigmentation [96]. Distinct from the response of low temperature, plants generate flavones to eliminate excess ROS at short-term high-temperature conditions in colorless plant organs. However, R2R3-MYB transcription factor CmMYB012 was found to respond to long-time high-temperature stress and inhibit flavonoid biosynthesis by directly regulating flavone synthase in the chrysanthemum [97]. In addition, CmMYB012 inhibited anthocyanin biosynthesis by inhibiting the expression of *CmCHS*, *CmDFR*, *CmANS,* and *CmUFGT* [97]. Moreover, in potatoes, StMYB44 negatively regulates anthocyanin biosynthesis at high temperatures in tuber flesh [98].

High temperature also induces the degradation of the HY5 protein in a COP1 activity-dependent manner and the degradation of HY5 derepresses the expression of *MYBL2*, which partially mediates the high-temperature repression of anthocyanin biosynthesis in *Arabidopsis* [99]. Furthermore, HY5 activates *miR858a*, which targets the anthocyanin repressor *MYBL2* and increases anthocyanin biosynthesis in *Arabidopsis* [100]. In apples, MdHY5 promotes anthocyanin accumulation by regulating the expression of the *MdMYB10* gene and downstream anthocyanin biosynthesis genes [25] and MdHY5 also inhibits the transcription of mdm-*miR858* which targets the transcription factor genes *MdMYB9* and *MdMYBPA1* to upregulate anthocyanin accumulation [101]. Conversely, miR858 negatively regulates anthocyanin biosynthesis in tomatoes by suppressing the expression of *SlMYB7-like* [102]. Therefore, the degradation of HY5 under high temperatures decreases anthocyanin biosynthesis through an opposite expression of *miR858*-*MYB* modules in different plants. Moreover, miR858 acts as a double-edged sword in anthocyanin regulation, and the activation or repression function depends on the target *MYB* genes [103].

Additionally, anthocyanin concentration depends on the counterbalance between its synthesis and degradation in plum fruit at high temperatures. The high temperature increased the concentration of hydrogen peroxide and the activity of class III peroxidase in the fruit. The hydrogen peroxide degraded anthocyanin, while class III peroxidase catalyzed hydrogen peroxide [104]. Moreover, ABA is an important positive regulator of the ripening and coloring of non-climacteric fruits while CYP707A (cytochrome P450 707A) and AOG (ABA β-glucosyltransferase) encode key enzymes in the catabolism and inactivation of ABA. High temperature delays the downregulation of *CYP707A* and *AOG* expression and leads to higher ABA catabolism and ABA inactivation in sweet cherry peel and leads to the reduction of anthocyanin accumulation [105].

Consequently, plants exposed to heat stress benefit from a decrease in anthocyanin because of the activation of negative factors, the degradation of positive regulators in anthocyanin biosynthesis, the high concentration of hydrogen peroxide, and the low level of ABA to degrade and reduce anthocyanin accumulation. The decreased anthocyanin accumulation under heat stress is caused by the transcriptional reduction of both early and late anthocyanin biosynthetic genes which are regulated by positive and negative regulators [100] and the acceleration of anthocyanin degradation [105].

However, during blueberry fruit maturation, VcmiRNA319 and ABA can target and regulate *VcMYBs* to be involved in anthocyanin biosynthesis [106]. In Asiatic hybrid lily flowers, high temperature suppressed miR828 accumulation and increased target *MYB12* transcription which enhanced anthocyanin pigmentation in flower tepals [107]. Therefore, high temperature usually negatively regulates anthocyanin accumulation and enhances heat stress tolerance by mediating the expression of heat stress-responsive genes (e.g., *HSFs* and *HSPs*) [108,109] rather than inducing the transcription of anthocyanin biosynthetic genes.

### 2.3. The Regulation of ncRNAs Involved in Salt Stress-Induced Anthocyanin Biosynthesis

Salt stress includes ionic, osmotic, and oxidative stress that affect the growth and development of plants. Excessive accumulation of Na^+^ and Cl^-^ in plant cells under salt stress breaks the osmotic balance, increases the concentration of ROS that are strongly oxidative, damage membrane proteins and membrane lipids, disrupt cell structure and function, damage biological macromolecules and enzyme systems [110]. However, ROS also transmit signals once plants encounter stress hazards via inducing cellular antioxidant mechanisms to remove excessive ROS. Through the MAPK pathway (mitogen-activated protein kinases), ROS activate anthocyanin regulatory factors (MYB, bHLH, WDR, bZIP) and regulate the expression of anthocyanin biosynthetic genes leading to anthocyanin accumulation. Then anthocyanins enhance cellular homeostasis and plant adaption to stress as antioxidants [111].

Plants have also evolved salt tolerance mechanisms through regulatory genes under salt stress. In torenia “Kauai Rose”, overexpression of anthocyanin regulatory transcription factors (*RsMYB1* or *B-Peru* + *mPAP1*) can alleviate salt stress-induced growth inhibition by reducing the salt stress-induced ROS and MDA contents [112]. In *Arabidopsis*, MYB3 functions as a transcriptional repressor for the regulation of lignin and anthocyanin biosynthesis under high salt conditions [113]. The *myb3* mutant plants exhibited high accumulation of lignin and anthocyanin and longer root growth in high NaCl conditions than wild-type plants and anthocyanin biosynthetic genes, such as *phenylalanine ammonia-lyase 1* (*PAL1*), *cinnamate 4-hydroxylase* (*C4H*), *catechol-O-methyltransferase* (*COMT*), *4-coumaric acid-CoA ligase* (*4CL*), *dihydroflavonol reductase* (*DFR*), and *leucoanthocyanidin dioxygenase* (*LDOX*) were also upregulated [113]. Moreover, the overexpression of *VvMYBA6* in *Arabidopsis*, significantly increased ABA and proline content, the activities of superoxide dismutase, peroxidase, catalase, and the accumulation of anthocyanins, and decreased the levels of H_2_O_2_ and malondialdehyde in response to salt stress [114]. Under salinity and high-light stress, MYB112 also positively regulates the expression of *PAP1*, *MYB114*, *MYB7,* and *MYB32* and promotes anthocyanin accumulation in *Arabidopsis* [115]. However, MYB7 and MYB32 have been reported to attenuate the transcriptional activity of the MBW complexes and repress anthocyanin biosynthesis in *Arabidopsis* [116] (Figure 4). This suggests the sophisticated regulation of environmental stress-induced anthocyanin biosynthesis.

Many miRNA target modules such as miR156-*SPL*, miR160-*ARF*, miR171-*SCL* (*SCARECROW-LIKE*), miR172-*AP2*, miR319-*TCP*, miR390-*TAS3*-*ARF*, miR393-*TIR1*/*AFB*, miR394-*LCR*, miR395-*APS/SULTR2;1*, miR396-*GRF*, miR397-*LAC*, miR398-*CSD*, miR399-*PHO2*, miR408-*LAC/PLANTACYANIN*, miR414-*FSD1*, miR528-*AAO/LAC*, miR535-*SPL*, and miRNAs such as miR402, miR417, miR1861h, miRNVL5 participate in plant salt stress responses by regulating hormonal signaling pathways and antioxidant system [117]. In *Arabidopsis* and *Oryza sativa*, both *35S::miR156* and *Ub::miR156* plants exhibited improved salt tolerance, and the accumulation of anthocyanin in transgenic *Arabidopsis* is determined by the miR156-*SPLs*-*DFR* pathway [118]. Anthocyanin can help plants respond to stressful environmental conditions and protect plants from damage [119].

However, in apple plants, *MIR156a* overexpression weakened salt resistance and *MdSPL13* (target of miR156) overexpression enhanced salt tolerance. Moreover, overexpression of *MdWRKY100*, the target of MdSPL13, also enhanced salt tolerance in apple [120]. Consequently, the miR156/*SPL* module regulates salt stress tolerance in apples by activating *MdWRKY100*. Thus, plants have developed various molecular mechanisms to adapt to salt environment stress, and the tolerance to salt depends on the target genes of miR156 in different plants.

Furthermore, in blueberry fruit coloration, the VcMIR156a-*VcSPL12* module regulates the anthocyanin accumulation by directly regulating the ethylene production pathway [121,122]. Ethylene has been reported to be a key regulator of salinity stress tolerance in plants via maintaining the homeostasis of Na^+^/K^+^, nutrients, reactive oxygen species (ROS), and cross-talk of ethylene signaling with other phytohormones [123]. Additionally, in rice and wheat, miR172/*IDS1* (*INDETERMINATE SPIKELET1*) regulatory module fine-tunes the expression of ROS-scavenging genes and ROS homeostasis during salt stress [124]. *MiR172*-overexpressing (172a-OE) transgenic rice plants enhanced salt tolerance and the “miR172/*IDS1-APXs/GPXs/CATs*-redox homeostasis” signaling pathway responded to the stress [124]. However, in apple plants, overexpression of miR172 reduces anthocyanin accumulation in various tissue types by suppressing the expression of an AP2 transcription factor that positively regulates *MdMYB10* [125] (Figure 4). As a result, overexpression miR172 in different plants displays diverse functions through varying miR172 target modules. Moreover, overexpression of lncERF024 in poplar enhances tolerance to salt stress [126].

### 2.4. The Transcription Factors and ncRNAs Involved in Drought Stress-Induced Anthocyanin Biosynthesis

Drought stress is one of the most common stresses in plants. When the water supply to the roots is limited or the transpiration rate is too high, the limited water supply leads to an imbalance between light capture and photosynthesis, allowing oxidative stress to occur. Drought stress induces reactive oxygen species (ROS), such as hydroxyl radicals (·OH), superoxide radicals (O_2_^•−^), and hydrogen peroxide (H_2_O_2_), and anthocyanins exhibit strong radical-scavenging activity avoiding excess ROS accumulation to cause cell death [127].

Under drought conditions, overexpression of *BoNAC019* from *Brassica oleracea* decreased the content of antioxidant enzymes and anthocyanin which conversely results in the accumulation of more reactive oxygen species (ROS). Moreover, the expression of antioxidant enzymatic genes, anthocyanin biosynthetic genes, and ABA signaling genes were also downregulated [128]. Moreover, in *Arabidopsis*, the loss-of-function of miR159 enhanced drought tolerance and hypersensitivity of seed germination to ABA, while *MYB33*, the target of miR159, acts upstream of ABI5 in the ABA signaling pathway to regulate drought response and seed germination in plants [129]. Furthermore, the overexpression of *MYB65* and *MYB101* in *Arabidopsis*, with a mutated recognition site in miR159, causes hypersensitivity to ABA and a relatively high tolerance to drought conditions [130]. However, in cotton, *GhMYB33* targeted by miR319c controls the transcription of *GhDFR1* to promote the accumulation of anthocyanin [131] (Figure 5). Therefore, in different plants, the same target gene can be regulated by distinct miRNAs and participate in different metabolic pathways.

The expression of Stu-miR159 decreased in response to drought treatment, while the expression of *StGAMyb-like* target genes increased with drought stress in potatoes [132]. Similarly, in tomatoes, the expression of sly-miR159 decreases in response to drought stress, and its target *SlMYB33* correlates with the induction of *SlP5CS* gene expression and accumulation of the osmoprotective compounds proline and putrescine [133]. Moreover, in mulberry, mul-miR159a plays a negative regulatory role in the biosynthesis of anthocyanins by targeting the Mul-*MYB33* gene [134]. Therefore, the decreasing expression of miR159 under drought stress might be related to the accumulation of anthocyanin in plants. Furthermore, in apple plants, MdMYB1 interacts with MdBT2, which plays a negative regulator role in the anthocyanin biosynthesis mediated by ABA, wounding, drought stress, and high light [135]. In addition, the interaction between MdERF38 and MdMYB1 enhances the binding of MdMYB1 to its target gene *MdDFR* and *MdUF3GT* and promotes the accumulation of drought-mediated anthocyanins [136]. Moreover, MdMYB1 binds to the *MBS* box of the *miR7125* promoter and promotes the expression of *miR7125* in apple fruit to increase anthocyanin biosynthesis through the regulation of the MdMYB16/MdMYB1-miR7125-*MdCCR* module [137]. Consequently, MdMYB1 might be the key factor involved in drought stress response and anthocyanin biosynthesis.

Additionally, in *Arabidopsis*, STTM165/166 promotes the expression of *HD-ZIP IIIs*, which also activates the expression of *ARFs* (AUXIN RESPONSE FACTORS) targeted by miR160 and shows tolerance to drought stress [138]. Moreover, miR160h-*ARF18* was identified as potentially controlling the accumulation of anthocyanins in poplar [5]. Furthermore, in apples, MdARF13 interacts with MdMYB10 and also binds to the promoter of *MdDFR*, which acts as a negative regulator of the anthocyanin metabolic pathway [139]. Also, overexpression of miR165 in *Arabidopsis* shows anthocyanin accumulation in the narrow cotyledons [140]. Thus, the interaction between miR160 and miR165/166 might be involved in the control of anthocyanin biosynthesis and drought tolerance of plants (Figure 5).

However, overexpression of osa-*MIR171* in rice exhibits a reduced tiller number and an increased flag leaf length compared to NT plants, which enhances the tolerance of drought stress through osa-miR171/SCL6 module regulating the expression of flavonoid biosynthesis genes [127]. Conversely, overexpression of *AtmiR858a* in tobacco decreases the expression of *NtMYB12* and regulates the biosynthesis of flavonoids leading to plants’ sensitivity toward drought stress. Simultaneously, the expression of target mimic for *miR858* (*MIMIC858*) in tobacco shows a short primary root length and enhances the expression of *NtMYB12* and flavonoids biosynthetic genes to confer drought tolerance [141]. Therefore, different miRNAs play important roles in regulating anthocyanin biosynthesis by targeting diverse transcription factor genes for drought stress resistance (Figure 5).

### 2.5. Long ncRNAs Related to Anthocyanin Synthesis

LncRNA can act as a regulator to affect the expression of genes involved in the anthocyanin biosynthetic pathway. For instance, *MdLNC499*, a long non-coding RNA regulated by MdWRKY1, induces the expression of *MdERF109*, which is involved in the light-induced anthocyanin synthesis pathway in apple fruit [49]. Moreover, MdLNC610, a positive regulator of *MdACO1* in ethylene biosynthesis, is also involved in the regulation of anthocyanin production induced by the high-light intensity in apple (*Malus domestica*) [48]. Additionally, LncRNAs also serve as precursors and endogenous target mimics (eTMs) of certain miRNAs to regulate the genes targeted by miRNAs. For instance, *MLNC3.2* and *MLNC4.6* function as eTMs for miR156a, and overexpression of the eTMs prevents cleavage of *SPL2-like* and *SPL33* by miR156a promoting light-induced anthocyanin accumulation in apple fruit [142]. Furthermore, *LNC1* and *LNC2* function as eTMs for miR156a and miR828a, which target *SPL9* and *MYB114* separately to regulate anthocyanin biosynthesis during sea buckthorn fruit ripening [47]. Moreover, *FRILAIR* acts as a noncanonical target mimic of miR397 to regulate the expression of its target gene *LAC11a* which promotes expressions of genes involved in the anthocyanin biosynthesis pathway during strawberry fruit ripening [88]. In addition, over-expressed miR397 delays fruit maturation, and the anthocyanin content also decreases [88]. Furthermore, in radish, overexpression of *LINC15957* increases anthocyanin accumulation and the expression of anthocyanin biosynthetic genes in leaves [143]. Many lncRNAs have been identified to be involved in anthocyanin biosynthesis through targeting miRNAs or directly affecting the expression of anthocyanin biosynthetic genes. The accumulation of anthocyanins in plants regulated by these ncRNAs is possibly related to the tolerance of plants to varied abiotic stresses.

## 3. Conclusions and Perspectives

Abiotic stress has extremely inhibited plant growth and crop productivity with the ongoing deterioration of the global climate and environment. However, plants have also evolved the defense mechanism to adapt to various stresses. Stresses induce the production of reactive oxygen species (ROS) and over-accumulation of ROS causes oxidative damage to plants. Simultaneously, anthocyanins are major bioactive compounds induced by abiotic stress as potent antioxidants to scavenge ROS [38]. Under cold stress, CBFs-dependent and -independent regulation of anthocyanin biosynthesis and *COR/LTI/RD/ERD* genes take roles in plant cold tolerance. However, the high temperature usually decreases anthocyanin accumulation and enhances tolerance to heat stress through regulating the expression of *HSFs* and *HSPs*. Moreover, salt stress induces MYB transcription factors and miRNAs involved in the regulation of anthocyanin and antioxidant enzymes for ROS clearance to increase stress tolerance. Furthermore, drought stress activates the pathway of anthocyanin biosynthesis, ABA, and auxin signaling to regulate the drought tolerance response. The species and roles of anthocyanins in ROS scavenging remain to be further studied. Given the critical roles that plant ncRNAs also play in various biological processes, further research is required to determine the relationship between ncRNA TF modules and anthocyanin synthesis-related structural genes and how they control anthocyanin biosynthesis when plants are subjected to abiotic stresses. The induction of anthocyanin synthesis by signaling factors associated with abiotic stresses also requires further study. Additionally, the construction of the LncRNA–miRNA–mRNA network is also not comprehensive and clear. In conclusion, with the progress of genetics and molecular biology, a clearer understanding of the biosynthesis and accumulation mechanisms of anthocyanins under abiotic stress is being achieved, which may provide a theoretical and practical basis for future crop breeding.

## Figures and Tables

**Figure 1 antioxidants-13-00055-f001:**
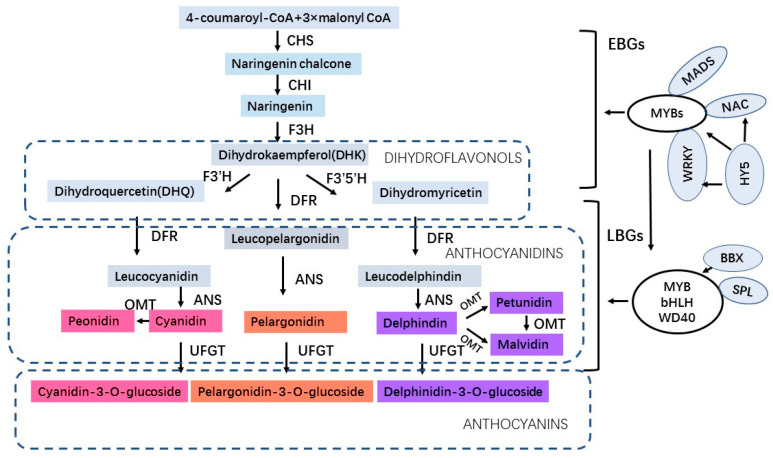
The regulatory pathway of anthocyanin biosynthesis.

**Figure 2 antioxidants-13-00055-f002:**
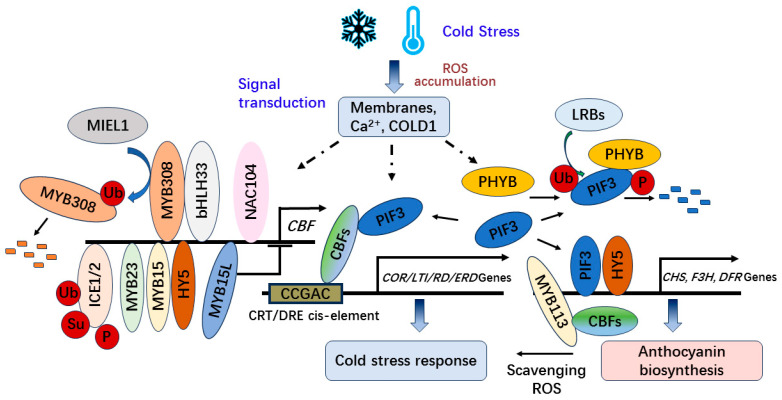
The CBF-dependent regulation of anthocyanin biosynthesis and cold tolerance under cold stress.

**Figure 3 antioxidants-13-00055-f003:**
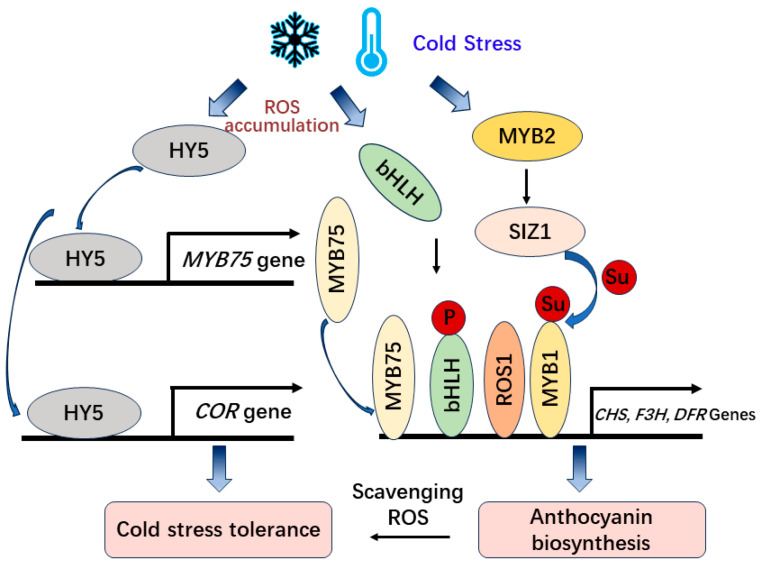
The CBF-independent regulation of anthocyanin biosynthesis and cold tolerance under cold stress.

**Figure 4 antioxidants-13-00055-f004:**
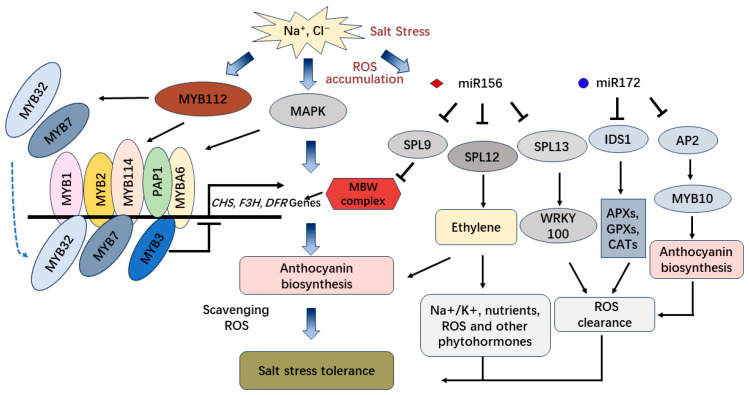
The regulation pathway of miRNAs and transcription factors involved in salt tolerance and anthocyanin biosynthesis.

**Figure 5 antioxidants-13-00055-f005:**
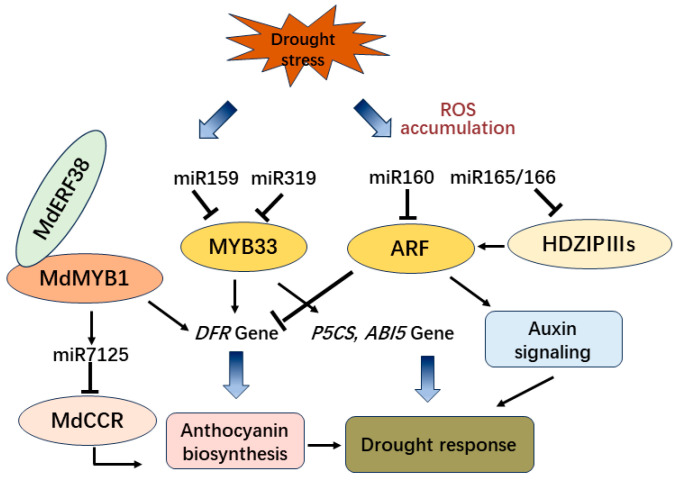
The regulation pathway of miRNAs and transcription factors involved in drought tolerance and anthocyanin biosynthesis.

**Table 1 antioxidants-13-00055-t001:** miRNAs and lncRNAs identified to take roles in anthocyanin accumulation and cold stress tolerance in plants.

Species	Non-Coding RNA	Targets/Downstream	Function	Methods	References
Sugarcane	MiR156	*ERD10B*, *ERD10C, ERD10D* and *LEA*	Cold tolerance and anthocyanin accumulation	Transient expression assay of OE-*miR156* in tobacco leaves	[81]
Chinese cedar	miR828/858	*MYBs*	Regulating needle discoloration in cold winters	Integrated transcriptome and miRNA analysis	[83]
Oryza sativa	MiR319b	*PCF6*, *TCP21*	Positively regulating cold tolerance	Overexpressing Osa-miR319b	[84]
Arabidopsis	miRJAW	*TCP3*	Negatively regulating anthocyanin biosynthesis	*p35S*::*mTCP3* and *p35S*::*TCP3SRDX*	[85]
Arabidopsis	MiR397	*LACs*, *CKB3*	Improving tolerance to cold stress	Overexpressing *miR397*	[87]
Strawberry	lncRNA FRILAIR	*MiR397*, *LAC11a*	Promoting fruit maturation and anthocyanin accumulation	*FRILAIR*, *LAC11a* overexpressing, *miR397* knockdown	[88]
Arabidopsis	MiR408	*LAC12* and *LAC13*	Increasing anthocyanin content	Artificial miRNA silences *MIR408*	[89]
Oryza sativa	MiR408	/	Improving tolerance to cold stress	Overexpressing *miR408*	[90]
Winter wheat	MiR398	*lncR9A*, *lncR117* and *lncR616*, *CSD1*	Improve cold resistance	*lncR9A* transferred Brachypodium distachyom	[91]
Peanut	miR398	*CHS*	Regulating anthocyanin biosynthesis	Transcriptome-metabolome joint analysis	[92]
Apple fruit	LNC499	*ERF109*	Anthocyanin accumulation	Transient *MdLNC4999* expression and *MdERF109* stable transformation	[49]
Trifoliate orange	/	*ERF109* *Prx1*	Cold tolerance	Overexpression and VIGS of *PtrERF109*	[93]
Apple plant	LNC610	*ACO1*	Anthocyanin accumulation	Overexpression of *MdACO1* and *MdLNC610*	[48]
Arabidopsis	CIL1	/	Positive response to cold stress	Knockdown *cil1* mutants	[95]

## Data Availability

Not applicable.

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
