# Peer review of "The ncRNAs Involved in the Regulation of Abiotic Stress-Induced Anthocyanin Biosynthesis in Plants"

_antioxidants, 2023, doi:10.3390/antiox13010055_

Round 1

Reviewer 1 Report

Comments and Suggestions for Authors

The review is interesting a provides plenty and useful information about a topic of major interest with many applications and outcomes. Nevertheless I have some suggestiond for improvement.

In page 2 there is a long paragraph with a description of the anthocyanin biosynthesis and regulation (lines 50-80). Aiming to make the paper more appealing for public not familiar with anthocyanins biosyntesis I would recommend to include a figure with a description of the bioynthesis and the know regulators, to help readers and to make a graphic introduction.

Minor points: 

Line 10: Spacing.

Line 27-28: climate change is a problem for all plants, not onlu for crop plants. It has also a great impact on antural environents.

So please, correct this sentence.

Comments on the Quality of English Language

English is fine, but I recommend an stilistic correction to improve the readibility.

Author Response

Reviewer 1

The review is interesting a provides plenty and useful information about a topic of major interest with many applications and outcomes. Nevertheless I have some suggestiond for improvement.

Thank you for your comments.

In page 2 there is a long paragraph with a description of the anthocyanin biosynthesis and regulation (lines 50-80). Aiming to make the paper more appealing for public not familiar with anthocyanins biosyntesis I would recommend to include a figure with a description of the bioynthesis and the know regulators, to help readers and to make a graphic introduction.

Thank you for your suggestions. We have added a figure about the pathway of anthocyanin biosynthesis in the paragraph.

Minor points:

Line 10: Spacing.

We have deleted the redundant space.

Line 27-28: climate change is a problem for all plants, not onlu for crop plants. It has also a great impact on antural environents.

So please, correct this sentence.

Thank you for your comments. We have revised the sentence to “With population growth and environmental deterioration, plants face more and more abiotic stresses.”

Reviewer 2 Report

Comments and Suggestions for Authors

The manuscript “The ncRNAs involved in the regulation of abiotic stress-induced anthocyanin biosynthesis in crop plants” summarized current knowledge about noncoding RNA (ncRNA) regulation of anthocyanin biosynthesis in various plant species. Authors discussed the micro RNA (miRNA) and long non-coding RNA (lncRNA) contribution against cold, salt and drought stresses through the anthocyanin biosynthesis. The paper focused on each environmental stress and ncRNA interaction and organized by the figures. However, some points should be corrected or improved.

1.     Some of the word font are different from the other part. Authors need to consist of the font through the paper. For example, line 2 “ncRNA” and line 37 “have” should be corrected.

2.     In line 63, Author wrote “N-induced anthocyanin”. Please clarify what N means.

3.     The same name miRNAs such as miR319 and miR408 were introduced in the paper. They were identified in different plant species including Arabidopsis and rice. Please describe whether the sequences of the same name miRNAs are well conserved in the plant species or not. It will help to understand the miRNA function.

Author Response

Reviewer 2

The manuscript “The ncRNAs involved in the regulation of abiotic stress-induced anthocyanin biosynthesis in crop plants” summarized current knowledge about noncoding RNA (ncRNA) regulation of anthocyanin biosynthesis in various plant species. Authors discussed the micro RNA (miRNA) and long non-coding RNA (lncRNA) contribution against cold, salt and drought stresses through the anthocyanin biosynthesis. The paper focused on each environmental stress and ncRNA interaction and organized by the figures. However, some points should be corrected or improved.

  1. Some of the word font are different from the other part. Authors need to consist of the font through the paper. For example, line 2 “ncRNA” and line 37 “have” should be corrected.

Thank you for your suggestions. We have checked the whole paper and revised the style of the font.

  1. In line 63, Author wrote “N-induced anthocyanin”. Please clarify what N means.

We have revised the sentence to “the low nitrogen (N)-induced anthocyanin accumulation”. N means nitrogen.

  1. The same name miRNAs such as miR319 and miR408 were introduced in the paper. They were identified in different plant species including Arabidopsis and rice. Please describe whether the sequences of the same name miRNAs are well conserved in the plant species or not. It will help to understand the miRNA function.

Thank you for your suggestions. MiR319 and miR408 are highly conserved among different species, which has been reported.

Li Y, Li C, Ding G, Jin Y. Evolution of MIR159/319 microRNA genes and their post-transcriptional regulatory link to siRNA pathways. BMC Evol Biol. 2011 May 12;11:122. doi: 10.1186/1471-2148-11-122. PMID: 21569383; PMCID: PMC3118147.

Guo, Y., Wang, S., Yu, K. et al. Manipulating microRNA miR408 enhances both biomass yield and saccharification efficiency in poplar. Nat Commun 14, 4285 (2023). https://doi.org/10.1038/s41467-023-39930-3

The sequences of mature miRNA are as follows

Ath-miR319a  154 - UUGGACUGAAGGGAGCUCCCU – 174

Ath-miR319b  152 - UUGGACUGAAGGGAGCUCCCU – 172

OsamiR319a    56 - AGCUGCCGAAUCAUCCAUUCA – 76

OsamiR319b   168 - UUGGACUGAAGGGUGCUCCC – 187

Ath-miR408   53 - ACAGGGAACAAGCAGAGCAUG – 73

Osa-miR408   31 - CAGGGAUGAGGCAGAGCAUGG – 51

We have revised the sentence in line 238 and line 252.

Reviewer 3 Report

Comments and Suggestions for Authors

Dear Authors,

I have reviewed your manuscript "The ncRNAs involved in the regulation of abiotic stress-induced anthocyanin biosynthesis in crop plants", submitted for publication in Antioxidants.

Your paper covers an interesting and emerging topic, which does not seem to be systematically covered in the existing literature so far. Moreover, the Authors give a comprehensive and thorough overview of the so-far published findings, making their paper a useful source of information and directions for anyone interested in the topic.

In its current form, the manuscript presents minor flaws, mostly related to the clarity of certain statements, extensive condensation of the text (which, at moments, makes it slightly challenging for the reader to follow), and minor language imperfections. Please follow my recommendations for improvement as listed below:

·         Manuscript title - Please consider deleting the word "crop" from the manuscript title. Although the majority of research covered by your review refers to crop plants, there is also fundamental research performed on non-crop species such as Arabidopsis.

·         Abstract - The Abstract is clear and well-written, I recommend several minor interventions:

o    line 15: please add the word "are" (...and respiration regulation ARE involved in plant adaptation to stress)

o    line 16: please revise: "anthocyanin biosynthesis IN RELATION TO abiotic stress reponse"

o    microRNAs (miRNAs) and long non-coding RNAs (lncRNAs) need to be fully spelled out within the Abstract text.

·         Introduction:

o    line 33: please replace "chemicals" with "messengers" or "regulators"

o    line 42-45: In this sentence, you imprecisely refer to anthocyanins and all of their precursors (anthocyanidins) collectively as "anthocyanins", which makes this sentence incorrect. Please thoroughly revise. Also, an additional figure, illustrating the biosynthetic pathway of anthocyanins, including their major molecular forms and the major enzymes involved in their biosynthesis and molecular modifications, might be considered as a useful addition to spare the reader from having to check other sources for a reminder of the anthocyanin biosynthesis pathway.

o    line 60-61: please replace "fruit plants" with "fruit crops"

o    line 66 (and elsewhere, please thoroughly check the whole manuscript): species names such as Cymbidium, should always be written in italic letters

o    line 68, 69, and elsewhere (line 231, please check the whole manuscript): plants overexpressing certain genes, are not "overexpressed", but "overexpressing" plants, please thoroughly revise

o    line 73: please replace "excessive" with another word ("extensive" might be suitable). Plants have evolved mechanisms of response to stress by progressively adapting through evolution, so their production of ROS cannot be called "excessive". Speaking of which, in this portion of text (lines 72-79) a more elaborated clarification would be needed to summarize how a high-damage-inducing event such as the production of ROS by the plant itself, can be beneficial for its adaptive response to stress.

o    line 77 and elsewhere, throughout the manuscript: At multiple points you start a sentence with "Then, ..." which incorrectly suggests that the following statement happens chronologically after what has been described in the previous sentence. Please thoroughly revise the manuscript language to remove this inaccuracy.

o    line 82-84: Please fully spell out the names of each category of RNAs listed here (including those which you had already spelled out in the Abstract). Most readers are familiar with some classes such as rRNA and miRNA, but some of these classes might not be familiar to a broad readership.

o    line 88: "are still need to be summarized" - please thoroughly check your English

·         Section 2:

o    line 111-112: "that give plants all kinds of colors" does not sound very scientific, please revise. Also, this entire paragraph (111-119) is mostly a repetition of what has already been said in the Introduction, please revise it to better announce what is coming in the follow-up (the summarization of specific regulatory pathways in response to each type of environmental stress).

o    line 125: please delete "corn". Corn is the literally the product of harvesting maize plants.

o    Section 2.1. in general, but also sections 2.2, 2.3, and 2.4: The text in these sections is at times difficult to follow because it is written as a single block of text. Please use the division of text into smaller paragraphs throughout the manuscript, to suggest the logical structure of the text to the reader.

o    line 147: please delete "in" and add "loss-of-function": "... and loss-of-function mutations hy5 or myb15..."

o    line 151-153: This sentence is unclearly formulated, and moreover, the Figure only shows phosphorylation but not ubiquitination. Please revise both the sentence and the Figure to reflect more clearly what is happening.

o    line 158-159: "which improve the contents of anthocyanin" - by which mechanism? Please provide a brief elaboration.

o    line 164: please delete "in apple (Malus domestica)" because you were already talking about the apple in the previous sentence. If you wanted to introduce the Latin name Malus domestica, you should do that at first mention, not here.

o    line 182: "independently of the CBF pathway" - please add a citation of Figure 2 to the end of this sentence to immediately direct the reader to look at the correct Figure

o    line 184: COR should be written in italic here because it refers to the gene family

o    line 191: please put the promoterS into plural here

o    line 192: This seems like a good example of where to start a new paragraph

o    line 211-212: Please revise: "to the increased expression of structural genes"

o    line 221-222: Overexpression OF MIR156 in tobacco leaves. Also: here in the text you mention the tobacco leaves, but the Table says that this research is about sugarcane. Please double-check and clarify or revise, as appropriate.

o    line 228: please revise: competitively interacting with PRODUCTION OF THE ANTHOCYANIN PIGMENTS1, to adequately suggest that you refer to a specific gen name here.

o    line 234-237: This sentence is unclear as to what is happening in rice and what is happening in Arabidopsis, please revise to make it more clear and to avoid "mixing apples and oranges"

o    line 241: with species-specific what? Effects?

o    line 253: please fully spell out CSD1

o    line 259: you have an unnecessary repetition of "biosynthesis" here. "Ethylene and anthocyanin biosynthesis" will do.

o    line 271: please delete "global". Individual plants can only respond to local changes in temperature, so global changes are irrelevant on the level of individual plants.

o    general remark about section 2.2: This section is the only section without visual summarization. Please consider adding a figure to this section to visually lead the reader through the text. Also please divide this section into paragraphs to suggest the logical structure of the text to the reader.

o    another general remark about this section: Of all the mentioned types of abiotic stress, heat is the only type of stress that generally leads to a decrease, rather than increase in anthocyanin content in plants. Is there an explanation of this phenomenon from an adaptive point of view? Why would plants exposed to heat stress benefit from a decrease in anthocyanin? This should be briefly elaborated within this section, citing appropriate literature sources.

o    line 293: please revise and add quotes to "double-edged sword" because "double-edged sword" is a metaphore here: miR858 acts as a "double-edged sword" in anthocyanin regulation

o    line 298-299: This sentence sounds nonsensical, please extensively revise, look at the original paper [105] for proper formulation of what is going on

o    line 315-319: The English language (the grammar structure of the sentence) needs to be revised here

o    line 329-330: "up-regulated by dissociation of MYB3 and Transparent Testa 8 (TT8) and Enhancer of Glabra 3 (EGL3)" - this sounds unclear. What does "up-regulated by dissociation" mean? Please revise, or very briefly elaborate to explain to the reader what kind of dissociation you are referring to here.

o    line 340-345: this sentence contains too many unnecessary repetitions of the word "module". You can just list all the modules and add the word "modules" at the end of listing them

o    line 366: please add "of": "overexpression OF miR172"

o    line 384-385: this sentence seems contradictory. Apparently, overexpression of BoNAC019 leads to both a decrease in antioxidant enzymes, and to the expression of their genes. Please double-check and revise the whole sentence.

o    Also please make sure that you cite Figure 4 when mentioning the first interaction that is shown in the Figure, to make sure that the reader is directed to look at the Figure as soon as reading the part of the text which is visualized in the Figure.

o    line 391: is the mutation in miRNA159 the cause of upregulation of these genes? Please be more clear about it. Also please write the gene names (MYB65 and MYB101) in italic, and please pay attention to always use the italic letters when you refer specifically to gene names (in contrast to proteins, which should be written in plain letters)

o    line 413: the text says that STTM165/166 promotes the expression of HD-ZIP IIIs, but the Figure shows this interaction as a downregulation. Please double-check and revise either the Figure or the text. Also please thoroughly check all the other Figures in the manuscript for possible similar inaccuracies, or inconsistencies between the Figures and the text

o    line 414: please fully spell AUXIN RESPONSE FACTORS at first mention

o    line 455-456: This sentence can be deleted since the same sentence already exists a few lines above

o    the Conclusions Section should be numbered as 3, not 5

o    Please write the Author Contributions in line with CRediT Taxonomy: https://credit.niso.org/

Comments on the Quality of English Language

The manuscript would benefit from the editing of English language. I have pointed out the main points for improvement within my review report.

Author Response

Reviewer 3

I have reviewed your manuscript "The ncRNAs involved in the regulation of abiotic stress-induced anthocyanin biosynthesis in crop plants", submitted for publication in Antioxidants.

Your paper covers an interesting and emerging topic, which does not seem to be systematically covered in the existing literature so far. Moreover, the Authors give a comprehensive and thorough overview of the so-far published findings, making their paper a useful source of information and directions for anyone interested in the topic.

In its current form, the manuscript presents minor flaws, mostly related to the clarity of certain statements, extensive condensation of the text (which, at moments, makes it slightly challenging for the reader to follow), and minor language imperfections. Please follow my recommendations for improvement as listed below:

Thank you for your valuable advices

  • Manuscript title- Please consider deleting the word "crop" from the manuscript title. Although the majority of research covered by your review refers to crop plants, there is also fundamental research performed on non-crop species such as Arabidopsis.

  We have revised the title and delete “crop”.

  • Abstract- The Abstract is clear and well-written, I recommend several minor interventions:

o    line 15: please add the word "are" (...and respiration regulation ARE involved in plant adaptation to stress)

o    line 16: please revise: "anthocyanin biosynthesis IN RELATION TO abiotic stress reponse"

o    microRNAs (miRNAs) and long non-coding RNAs (lncRNAs) need to be fully spelled out within the Abstract text.

Thank you. We have revised the abstract.

  • Introduction:
  • line 33: please replace "chemicals" with "messengers" or "regulators"

We have replaced "chemicals" with "messengers".

  • line 42-45: In this sentence, you imprecisely refer to anthocyanins and all of their precursors (anthocyanidins) collectively as "anthocyanins", which makes this sentence incorrect. Please thoroughly revise.Also, an additional figure, illustrating the biosynthetic pathway of anthocyanins, including their major molecular forms and the major enzymes involved in their biosynthesis and molecular modifications, might be considered as a useful addition to spare the reader from having to check other sources for a reminder of the anthocyanin biosynthesis pathway.

Thank you for your suggestions. We have added the figure about the pathway of anthocyanin biosynthesis..

  • line 60-61: please replace "fruit plants" with "fruit crops"

We have replaced "fruit plants" with "fruit crops".

  • line 66 (and elsewhere, please thoroughly check the whole manuscript): species namessuch as Cymbidiumshould always be written in italic letters

We have revised the species names in italic letters in the whole manuscript.

  • line 68, 69, and elsewhere (line 231, please check the whole manuscript): plants overexpressing certain genes, are not "overexpressed", but "overexpressing" plants, please thoroughly revise

Thank you. We have checked the whole manuscript and revised it to overexpressing plants.

  • line 73: please replace "excessive" with another word ("extensive"might be suitable). Plants have evolved mechanisms of response to stress by progressively adapting through evolution, so their production of ROS cannot be called "excessive". Speaking of which, in this portion of text (lines 72-79) a more elaborated clarification would be needed to summarize how a high-damage-inducing event such as the production of ROS by the plant itself, can be beneficial for its adaptive response to stress.

Thank you for your suggestions. We have revised the sentence of the reason for ROS production. L76-78.

  • line 77 and elsewhere, throughout the manuscript: At multiple points you start a sentence with "Then, ..."which incorrectly suggests that the following statement happens chronologically after what has been described in the previous sentence. Please thoroughly revise the manuscript language to remove this inaccuracy.

Thank you. We have revised the expression of “then” in the whole manuscript.

  • line 82-84: Please fully spell out the names of each category of RNAslisted here (including those which you had already spelled out in the Abstract). Most readers are familiar with some classes such as rRNA and miRNA, but some of these classes might not be familiar to a broad readership.

Thank you. We have revised the name of these RNAs.

  • line 88: "are still need to be summarized" - please thoroughly check your English

Thank you. We have revised it.

  • Section 2:
  • line 111-112: "that give plants all kinds of colors" does not sound very scientific, please revise. Also, this entire paragraph (111-119) is mostly a repetition of what has already been said in the Introduction, please revise it to better announce what is coming in the follow-up(the summarization of specific regulatory pathways in response to each type of environmental stress).

Thank you for your suggestion. The content is repetition with the introduction. We delete the paragraph.

  • line 125: please delete "corn". Corn is the literally the product of harvesting maize plants.

We have deleted “corn”.

  • Section 2.1. in general, but also sections 2.2, 2.3, and 2.4: The text in these sections is at times difficult to follow because it is written as a single block of text. Please use the division of text into smaller paragraphs throughout the manuscript, to suggest the logical structure of the textto the reader.

Thank you for your suggestion. We have divided the text into smaller paragraphs.

  • line 147: please delete "in" and add "loss-of-function": "... and loss-of-function mutationshy5 or myb15..."

Thank you. We have revised it.

  • line 151-153: This sentence is unclearly formulated, and moreover, the Figure only shows phosphorylation but not ubiquitination. Please revise both the sentence and the Figure to reflect more clearly what is happening.

We have revised the sentence and the figure.

  • line 158-159: "which improve the contents of anthocyanin" - by which mechanism?Please provide a brief elaboration.

We have revised the sentence to clarify the reason of anthocyanin biosynthesis.

  • line 164: please delete "in apple (Malus domestica)" because you were already talking about the apple in the previous sentence. If you wanted to introduce the Latin name Malus domestica, you should do that at first mention, not here.

We have deleted "in apple (Malus domestica)" in the sentence. L170

  • line 182: "independently of the CBF pathway" - please add a citation of Figure 2to the end of this sentence to immediately direct the reader to look at the correct Figure

Thank you. We have added the citation of figure.

  • line 184: CORshould be written in italic here because it refers to the gene family

Thank you. We have revised it.

  • line 191: please put the promoterSinto plural here

Thank you. We have revised it.

  • line 192: This seems like a good example of where to start a new paragraph

Thank you. We have started a new paragraph.

  • line 211-212: Please revise: "to the increased expressionof structural genes"

Thank you. We have revised it.

  • line 221-222: Overexpression OFMIR156 in tobacco leaves. Also: here in the text you mention the tobacco leaves, but the Table says that this research is about sugarcane. Please double-check and clarify or revise, as appropriate.

We have checked and revised it.

  • line 228: please revise: competitively interacting with PRODUCTION OF THE ANTHOCYANIN PIGMENTS1, to adequately suggest that you refer to a specific gen name here.

We have revised it.L235

  • line 234-237: This sentence is unclear as to what is happening in rice and what is happening in Arabidopsis, please revise to make it more clear and to avoid "mixing apples and oranges"

Thank you. We have revised the sentence L241-245.

  • line 241: with species-specific what? Effects?

Thank you. We have revised it to species specific features.L249

  • line 253: please fully spell out CSD1

We have revised the full name of CSD1. L262

  • line 259: you have an unnecessary repetition of "biosynthesis" here. "Ethylene and anthocyanin biosynthesis" will do.

Thank you. We have deleted it.

  • line 271: please delete "global". Individual plants can only respond to local changes in temperature, so global changes are irrelevant on the level of individual plants.

Thank you. We have deleted it.

  • general remark about section 2.2: This section is the only section without visual summarization. Please consider adding a figure to this section to visually lead the reader through the text. Also please divide this section into paragraphs to suggest the logical structure of the text to the reader.

In section 2.2, High temperature usually negatively regulates anthocyanin accumulation and the research about anthocyanin and ROS under high temperature is rare, so we do not add the figure. However, we divide this section into four paragraphs.

  • another general remark about this section: Of all the mentioned types of abiotic stress, heat is the only type of stress that generally leads to a decrease, rather than increase in anthocyanin content in plants. Is there an explanation of this phenomenon from an adaptive point of view? Why would plants exposed to heat stress benefit from a decrease in anthocyanin? This should be briefly elaborated within this section, citing appropriate literature sources.

Thank you for your suggestions. We have added one paragraph to explain the reason.L313-319

  • line 293: please revise and add quotes to "double-edged sword" because "double-edged sword" is a metaphore here: miR858 acts as a "double-edged sword" in anthocyanin regulation

Thank you for your suggestions. We have revised it. L307-309

  • line 298-299: This sentence sounds nonsensical, please extensively revise, look at the original paper [105] for proper formulation of what is going on

Thank you for your suggestions. We have revised it. L311-316

  • line 315-319: The English language (the grammar structure of the sentence) needs to be revised here

We have revised the sentences.L337-343

  • line 329-330: "up-regulated by dissociation of MYB3 and Transparent Testa 8 (TT8) and Enhancer of Glabra 3 (EGL3)" - this sounds unclear. What does "up-regulated by dissociation" mean? Please revise, or very briefly elaborate to explain to the reader what kind of dissociation you are referring to here.

We have deleted “by dissociation of MYB3 and Transparent Testa 8 (TT8) and Enhancer of Glabra 3 (EGL3)” in the sentence. L354

  • line 340-345: this sentence contains too many unnecessary repetitions of the word "module". You can just list all the modules and add the word "modules" at the end of listing them

Thank you for your suggestions. We have revised it. L364-369

  • line 366: please add "of": "overexpression OFmiR172"

We have added “of” before miR172. L390

  • line 384-385: this sentence seems contradictory. Apparently, overexpression of BoNAC019leads to both a decrease in antioxidant enzymes, and to the expression of their genes. Please double-check and revise the whole sentence.

We have revised the whole sentence. L408-412

  • Also please make sure that you cite Figure 4 when mentioning the first interaction that is shown in the Figure, to make sure that the reader is directed to look at the Figure as soon as reading the part of the text which is visualized in the Figure.

We have mentioned the figure 5 in L419.

  • line 391: is the mutation in miRNA159 the cause of upregulation of these genes?Please be more clear about it. Also please write the gene names (MYB65 and MYB101) in italic, and please pay attention to always use the italic letters when you refer specifically to gene names (in contrast to proteins, which should be written in plain letters)

Thank you. We have revised the sentence. L415-416.

  • line 413: the text says that STTM165/166 promotes the expression of HD-ZIP IIIs, but the Figure shows this interaction as a downregulation. Please double-check and revise either the Figure or the text. Also please thoroughly check all the other Figures in the manuscript for possible similar inaccuracies, or inconsistencies between the Figures and the text

We have checked the text and figure. STTM165/166 means “knock-down” or “silence” the function of miR165/166.

  • line 414: please fully spell AUXIN RESPONSE FACTORSat first mention

We have revised the full name of ARFs.L440

  • line 455-456: This sentence can be deleted since the same sentence already exists a few lines above

Thank you. We have deleted the sentence.

  • the Conclusions Section should be numbered as 3, not 5

Thank you. We have revised it.

  • Please write the Author Contributions in line with CRediT Taxonomy: https://credit.niso.org/

We have revised it.